# Rapid and Visual Screening of Virus Infection in Sugar Beets Through Polerovirus-Induced Gene Silencing

**DOI:** 10.3390/v16121823

**Published:** 2024-11-23

**Authors:** Heemee Devi Bunwaree, Elodie Klein, Guillaume Saubeau, Bruno Desprez, Véronique Ziegler-Graff, David Gilmer

**Affiliations:** 1Institut de Biologie Moléculaire des Plantes, CNRS-UPR 2357, Université de Strasbourg, 67000 Strasbourg, France; 2Florimond Desprez, 3 rue Florimond Desprez, 59242 Cappelle-en-Pévèle, France

**Keywords:** polerovirus, beet mild yellowing virus (BMYV), *Beta vulgaris*, viral yellows (VY), virus-induced gene silencing (VIGS), visual screening, agroinfiltration, small interfering RNA (siRNA)

## Abstract

Since the ban of neonicotinoid insecticides in the European Union, sugar beet production is threatened by outbreaks of virus yellows (VY) disease, caused by several aphid-transmitted viruses, including the polerovirus beet mild yellowing virus (BMYV). As the symptoms induced may vary depending on multiple infections and other stresses, there is an urgent need for fast screening tests to evaluate resistance/tolerance traits in sugar beet accessions. To address this issue, we exploited the virus-induced gene silencing (VIGS) system, by introducing a fragment of a *Beta vulgaris* gene involved in chlorophyll synthesis in the BMYV genome. This recombinant virus was able to generate early clear vein chlorosis symptoms in infected sugar beets, allowing easy and rapid visual discernment of infected plants across five sugar beet lines. The recombinant virus displayed similar infectivity as the wild-type, and the insert remained stable within the viral progeny. We demonstrated that the percentage of VIGS-symptomatic plants was representative of the infection rate of each evaluated line, and depending on the susceptibility of the line to BMYV infection, VIGS symptoms may last over months. Our work provides a polerovirus-based VIGS system adapted to sugar beet crop allowing visual and rapid large-scale screens for resistance or functional genomic studies.

## 1. Introduction

Sugar beet (*Beta vulgaris* subsp. *vulgaris* L.), cultivated for its saccharose-rich root, is the second largest source of the world’s sugar production and holds significant economic importance in Europe, the leading supplier [1]. Before the use of the neonicotinoid insecticides, infection with virus yellows (VY) could lead to losses in sugar yield between 18 and 47% depending on the responsible virus [2]. Since the early 1990s, this class of insecticides allowed effective control of the vector and therefore of the disease. Nevertheless, the ban of neonicotinoid in the European Union has severely constrained sugar beet production by important outbreaks of aphid-transmissible virus yellows (VY) in most European countries [2]. In the face of these recurrent epidemics, the development of resistant cultivars against VY is becoming increasingly urgent to sustain the sugar industry. However, identifying resistances from sugar beet accessions, which is the key step of the process, poses a great challenge for breeders.

In Europe, VY disease in sugar beets is caused by a complex of four aphid-transmissible viruses, namely the Beet yellows virus (BYV) of the genus *Closterovirus*, the Beet mosaic virus (BtMV) of the genus *Potyvirus*, the Beet mild yellowing virus (BMYV) and the Beet chlorosis virus (BChV), both belonging to the genus *Polerovirus* [2]. In addition to reduced taproot weight, VY is initially characterised by various intensities of yellowing of leaves, from yellowish destaining of the older leaves to reddish necrosis for BYV, yellow to orange leaf discoloration causing premature foliage death for poleroviruses, and yellowish speckles and leaf malformations for BtMV [2]. Such diversities in leaf symptoms account for the difficulty of the traditional evaluation for resistances relying primarily on expert scoring. However, visually discriminating non-infected plants from infected plants can be biased by mixed viral infections, other biotic or abiotic stresses [1], and conversely, some infected plants may show no symptoms at all. For accurate evaluation, serological or molecular analyses are often required, which can be time-consuming, costly and labour-intensive, making them incompatible for large scale screenings.

The polerovirus genome consists of a single-stranded positive sense RNA of around 5.7 kb, protected by a virus genome-linked protein (VPg) at its 5′ untranslated region (UTR), enclosed within isometric particles of 23 to 25 nm in diameter. The genome is organised into at least seven overlapping open reading frames (ORFs), expressed from genomic and subgenomic RNAs. Canonical and non-canonical translation mechanisms are implemented to produce proteins essential for systemic infection [3]. Due to the compact nature of the viral genome organisation and the packaging constraints into isometric virions, poleroviruses are recalcitrant to the integration of reporter genes for virus tracking [4,5]. Insertion of foreign sequences into the viral genome can affect its stability leading to deletions in the recombinant virus [6] or reduce the viral infectivity by compromising a series of intricate protein interactions essential for viral replication or systemic movement [7,8]. Nonetheless, the study undertaken by Bortolamiol-Bécet et al. [9] demonstrated that despite the aforementioned constraints, a polerovirus genome can be engineered as a virus-induced gene silencing (VIGS) tool to provoke noticeable specific vascular yellowing symptoms in the laboratory plant model *Arabidopsis thaliana*.

Over the past year, VIGS has emerged as a powerful tool for gene function studies in crop plants [10]. VIGS enables the rapid knockdown of host transcripts, without the need for stable transformation. A recombinant virus carrying a short sequence of the host target gene is introduced into the plant, through either mechanical inoculation or *Agrobacterium tumefaciens*-mediated inoculation, allowing the establishment of the viral infection [11,12]. During viral replication, double-stranded RNAs (dsRNA) are produced in the cytoplasm and trigger post-transcriptional gene silencing (PTGS) in the plant. Dicer proteins recognise and cleave the viral dsRNAs into small interfering RNAs (siRNAs) corresponding to the viral genome including the host gene sequence. The guide strands of these siRNAs associate with Argonaute proteins and are loaded into the RNA-induced silencing complex (RISC) for sequence-specific RNA degradation of complementary transcripts. Hence, by introducing an 81 base pairs (bp) sense fragment of the endogenous *CHLI1* gene, involved in chlorophyll synthesis in the 3′ UTR of the genome of the turnip yellows virus (TuYV), Bortolamiol-Bécet et al. [9] generated a recombinant polerovirus, capable of silencing the target gene *CHLI1*, which could be visualised as a robust vein-clearing phenotype, with the naked eye, and lasted up to at least 28 days post-inoculation in *A. thaliana*. The study also showed that the recombinant polerovirus was as infectious as the wild-type virus. An 81 bp sense insert was more stable than a hairpin-based 42 nts fragment within the viral progeny.

To facilitate the screening process in pre-breeding biotests in sugar beet, this work aimed to develop a molecular tool that enables fast and reliable visual identification of viral infection, particularly for the most reported member of VY—the beet mild yellowing virus [13]. A recombinant BMYV-based vector was generated by introducing a fragment of the *B. vulgaris CHLI* gene into the viral genome. Infectivity of the recombinant virus and its capacity of inducing visual symptoms were evaluated in five sugar beet lines. Finally, we investigated the temporal infection and silencing patterns on the BMYV-based VIGS in sugar beets.

## 2. Materials and Methods

### 2.1. Plant Material

Five *Beta vulgaris* lines were used in this experiment, which for confidential reasons are referred as A, B, C, D, and E. The seeds were sown in soil and three weeks later, seedlings were individually transplanted into nutrient-rich surfinia soil (HAWITA Gruppe GmbH, Vechta, Germany). The plants were then grown under greenhouse conditions at 22 °C for 16 h under high-pressure sodium light and 18 °C for a period of 8 h in darkness.

### 2.2. Construction of Recombinant BMYV-SUL

To develop a BMYV-derived VIGS vector, the full-length infectious 35S promoter-driven BMYV cDNA clone [isolate EK, [14]], constructed in pBluescript^®^ (pBS) vector, was used. A 70 bp fragment of endogenous gene, *CHLI* (GenBank accession XM_010688043, position 1268–1337 bp), referred to as SUL, was inserted at position 5614 of the viral cDNA clone. The pBluescript^®^ backbone and the BMYV cDNA were amplified by PCR, with the primer pairs SUL-pBS_F and 35S_R and M13_F and SUL-BMYV_R, respectively (Appendix A). The two PCR fragments were designed to have an overlap of 103 nucleotides at the M13 sequence of the pBS backbone and 40 nucleotides at the 3′ untranslated region of the viral cDNA, where the SUL fragment was inserted. The pBS PCR fragment and the BMYV PCR fragment were assembled at 50 °C for 1 h, by Gibson assembly method, using the *NEBuilderHiFi DNA Assembly Cloning Kit* (New England Biolabs, Inc., Ipswich, MA, USA). The sequence of the recombinant pBS-BMYV-SUL construct was controlled and further subcloned into binary vector pBIN19, by restriction enzymes *Kpn*I and *Spe*I, for transformation into electrocompetent *Agrobacterium tumefaciens*, strain GV3101.

### 2.3. Viral Infection of Beta Vulgaris by Agroinoculation

Four weeks’ seedlings were agroinoculated with either the full-length infectious cDNA clone of the wild-type BMYV [isolate EK] or the recombinant BMYV-SUL. The transformed *A. tumefaciens*, harbouring the binary plasmid containing the wild-type or recombinant BMYV genome, were grown overnight in LB medium supplemented with 100 µM kanamycin, 100 µM rifampicin, 10 mM MES pH 5.6, and 150 µM acetosyringone at 28 °C, in a shaker at 200 rpm. The agrobacterial cells were centrifuged at 5000 rpm for 20 min and resuspended in infiltration medium (10 mM MgCl_2_, 10 mM MES pH 5.6, and 150 µM acetosyringone) to be adjusted to a final optical density at 600 nm (OD_600nm_) of 0.8. One or two tiny holes were delicately made at the tip and on the lower surface of the cotyledons or the first pair of leaves by using a 1.2 mm needle. The suspension was then infiltrated to the leaves, by gently pressing a needleless syringe, as previously described by English et al. [15].

### 2.4. Sample and ELISA Standards Preparation for ELISA

Leaf samples of five discs (approximately 110 mg) were randomly harvested from newly developed leaves of each plant, using a hole punch cutter. The discs were collected into Precellys^®^ tubes containing 5 glass beads (diameter 1.7–2.1 mm, Carl Roth, Karlsruhe, Germany) and ground to powder with liquid nitrogen by the Precellys^®^ Evolution tissue homogeniser (Bertin Technologies, Montigny-le-Bretonneux, France). Then, 400 µL of extraction buffer (137 mM NaCl, 8 mM Na_2_HPO_4_, 12 H_2_O, 2.7 mM KCl, 1.5 mM KH_2_PO_4_, 0.05% (*v*/*v*) Tween 20, 2% polyvinylpyrrolidone, pH 7.4) was added before another grinding by the Precellys^®^ lyser to extract plant sap for virus detection by triple-antibody sandwich (TAS-) ELISA.

To establish a reference standard for all ELISA experiments, ten 8-weeks-old sugar beet plants of line E, infected with BMYV-SUL, were pooled by grinding all the leaves into a fine powder. An amount of 110 mg of the ground sample was aliquoted in Precellys^®^ tubes containing 5 glass beads. The tubes were stored at −80 °C and per ELISA microtiter plate, one tube was used as standard sample (S_0_). For quantification purposes, a serial dilution by two-fold of the given sample in extraction buffer was carried out (S_0_ to S_10_) and loaded on the plate.

### 2.5. Calibration Curve, Virus Detection, and Semi-Quantification by TAS-ELISA

A calibration standard curve of viral concentration versus ELISA absorbance values was generated by using a serial dilution of purified virus. Purified BMYV virions were extracted from BMYV-EK *Montia perfoliata*-infected plants [16] and obtained from V. Brault (INRAE, Colmar, France). An amount of 4.5 ng/µL of purified BMYV (V_0_) was successively diluted by two-fold, by adding an equal volume of extraction buffer each time, and this step was repeated twelve times until a final concentration of 1 pg/µL (V_12_) was obtained. This serial dilution was then analysed by TAS-ELISA by adding 100 µL of the thirteen samples (V_0_ to V_12_) on a microtiter plate.

TAS-ELISA assays were performed by using anti-BWYV antibodies, a polerovirus closely related to BMYV (DSMZ, Braunschweig, Germany). The plates were coated by polyclonal IgG-BWYV antibodies according to manufacturer’s instructions. The extracted plant sap was centrifuged at 4000× *g* for 4 min at 4 °C and 100 µL of the supernatant sap was deposited on the same plate as the previously mentioned serial dilutions of either purified BMYV (V_0_) or standard sample (S_0_). After 3 h incubation with monoclonal anti-BWYV antibodies and 1 h with secondary alkaline phosphatase coupled antibodies, the OD of the colorimetric reaction was measured 30 min and 60 min after incubation, using a FLUOstar^®^ Omega microplate reader (BMG LABTECH, Champigny-sur-Marne, France) at a wavelength of 405 nm. The threshold for assay positivity was set at twice the mean OD_405nm_ value of the non-inoculated controls as described by Armand et al. [17].

### 2.6. Statistical Modelling

To establish the relationship between the virus concentration and OD values, a standard calibration curve of the serial dilution of purified BMYV was generated by using a five-parameter logistic (5PL) regression model, adapted to dual logarithm calculations, as described by Lange et al. [18]. The adapted 5PL model was fit to the serial dilution data points according to the following equation:$$\hat{{OD}_{i}}=\tilde{bc}+\frac{tl-\tilde{bc}}{{\left.\left[1+2^{S\;\cdot \left(ld\left(l\right)-ld\left(C_{i}\right)\right)}\right.\right]}^{A}}$$
where ${\hat{OD}}_{i}$ is the OD value of a given sample, *i* within the serial dilution, *C_i_* is the relative virus concentration of the sample *i*. *A* describes the asymmetry of the curve and was set to be one (*A* = 1). *I* is the relative virus concentration (*C_i_*) at the inflection point if *A* = 1, and *S* the slope at the inflection point, which was also set to be one (*S* = 1). The median of the buffer controls of each respective microtiter plate, $\tilde{bc}$ was set as the bottom asymptote of the curve and ideally, the top asymptote should correspond to the technical limit of the spectrophotometer, *tl*, that is an OD_405nm_ of 4.

### 2.7. Non-Linear Transformation of OD Values

Before using the standard sample S_0_ as reference for all upcoming ELISA experiments, six aliquots were serially diluted by two-fold and analysed in duplicates on three microtiter plates. On each microtiter plate, a serial dilution of purified virus was equally added to plot the calibration curve. The OD values of each sample, measured 30 min, 60 min and/or 180 min, after addition of substrate, were then transformed by employing the inverse of the previously described logistic regression model, fit to the serial dilution of purified virus, as per the following equation:$$ld(\hat{C_{i}})=\left(\frac{I}{{\left[{\left(\frac{tl-\tilde{bc}}{{OD}_{i}-\tilde{bc}}\right)}^{\frac{1}{A}}-1\right]}^{\frac{1}{S}}}\right)$$
where ld (${\hat{C}}_{i}$) is the dual logarithmic relative virus concentration, transformed from the *i*th OD value, OD*_i_*. The viral concentration of each sample (S_0_ to S_12_) was thus estimated by calculating the mean of the transformed OD values at 30 min and 60 min measurements [19]. This allowed to establish the relationship between OD values and virus concentrations, in its dual logarithmic form, from plant material S_0_ instead of purified virus V_0_ for non-linear transformation of OD values. Thus, for each ELISA experiment, a serial dilution of S_0_ is added to the microtiter plate to generate the standard curve and estimate the virus concentration of each plant from its OD value by the inverse-adapted 5PL logistic regression model.

### 2.8. Statistical Analysis

Statistical analyses were performed using R version 4.1.2 [20]. To statistically compare the level of infection of plants under different conditions, a Kruskal–Wallis test was performed by considering the medians of the estimated virus concentrations of the different groups, followed by a Dunn’s test for pairwise comparisons and Bonferroni correction. All tests were performed using a significance level of α = 0.05.

### 2.9. Viral Progeny Analysis by RT-PCR and Sanger Sequencing

Total RNA was extracted from 100 mg of *B. vulgaris* leaves by using either “polysomes” buffer [21] or TRIzol^TM^ reagent, followed by phenol and phenol/chloroform purification and ethanol precipitation [22,23]. Reverse transcription of 1 μg of extracted RNA was performed using the oligonucleotide BMYV_CT_rev2019 hybridising to the 3′ end of the viral genome (Appendix A) and the SuperScript IV Reverse Transcriptase (ThermoFischer Scientific™, Waltham, MA, USA) according to manufacturer’s recommendations. The cDNA was amplified by PCR, using Phusion™ High-Fidelity DNA Polymerase Master Mix 2X (ThermoFischer Scientific™), according to the manufacturer’s instructions. The sense primer BMYV_5300_F and antisense primer BMYV_CT_rev2021 were used to generate a PCR fragment corresponding to the 3′ end of BMYV RNA containing the SUL insert (Appendix A). The PCR products were purified from agarose gel and extracted by phenol and phenol-chloroform and precipitated by ethanol before being sequenced.

### 2.10. Viral RNA Detection by Northern Blot

Total RNA was extracted from 100 mg of *B. vulgaris* leaves with TRIzol^TM^ reagent and 5 µg of RNA extracts were mixed with 20 µL of RNA denaturing buffer (1X HEPES, 50% deionised formamide, 6% formaldehyde, and 0.006% bromophenol blue in 50% glycerol). The samples were heated to 65 °C for 10 min, cooled on ice and loaded onto a 1.5% agarose-1X TBE (Tris-Borate-EDTA) gel. After electrophoresis at 70 V in 1X TBE buffer, RNA samples were transferred overnight by capillarity onto a positively charged nylon membrane (Amersham Hybond™-NX, GE HealthCare, Chicago, IL, USA) and UV-crosslinked using a Stratalinker^®^. The membrane was stained with methylene blue (0.02% in 0.3 M sodium acetate, pH 5.2) to visualise RNA transfer, washed with water and destained by three washes in 0.2X saline-sodium citrate (SSC) buffer, 0.1% SDS. The membrane was then blocked with PerfectHyb™ Plus Hybridisation Buffer (Sigma-Aldrich, Steinheim, Germany), followed by overnight hybridisation at 55 °C with a DNA probe targeting genomic and subgenomic BMYV RNAs. The probe was prepared by amplifying the region coding for protein P3P5 using sense primer BMYV_NB_F and antisense primer BMYV_NB_R (Appendix A), with BMYV-EK cDNA as the template. The PCR product was radiolabelled with [α-32P]-dCTP, using the Prime-a-Gene^®^ Labelling System (Promega Corporation, Madison, WI, USA), according to the manufacturer’s instructions. The membrane was washed three times with 2X SSC, 0.1% SDS, followed by two washes with 0.2X SSC, 0.1% SDS at 65 °C. The membrane was exposed to a phosphor-imager screen and analysed with a phosphor-imager (Amersham Typhoon, GE HealthCare).

### 2.11. Small RNA Detection by Northern Blot

Forty micrograms of TRIzol-extracted RNA samples were dehydrated in a SpeedVac concentrator (SPD111V, ThermoFischer Scientific™), resuspended in 10 µL of small RNA-loading dye [22] and heated at 95 °C for 3 min. The RNA samples were loaded onto a polyacrylamide gel, electrophoresed and transferred to an Hybond-N+ nylon membrane (Amersham Hybond^TM^-N+) according to Böhrer et al. [22]. A DNA probe was prepared by amplifying the 70 bp SUL fragment, using the sense primer SUL-F and antisense primer SUL-R (Appendix A), and BMYV-SUL cDNA as template. The PCR product was radiolabelled by Klenow fragment enzyme (ThermoFischer Scientific™) and incubated with the membrane as described [22]. The membrane was exposed to a phosphor-imager screen and the signal was detected using a phosphor-imager (Amersham Typhoon, GE HealthCare). After stripping, the membrane was hybridised with the universal probe miR160 radiolabelled with [γ-32P]-dATP [22]. The membrane was finally exposed to a phosphor-imager screen.

## 3. Results

### 3.1. Virus-Induced Gene Silencing Mediated by BMYV in Sugar Beet

Bortolamiol-Bécet et al. [9] reported that poleroviruses can be effectively used as a virus-induced gene silencing (VIGS) vector to elicit a prominent yellow vein-clearing phenotype in *A. thaliana*, without compromising the virus’s infectivity. In order to assess whether this technique can be effectively applied to sugar beet, the BMYV cDNA clone [14] was accordingly modified by adding at the 3′ non-coding end of the viral genome a fragment of the endogenous *B. vulgaris CHLI* gene (SUL), a commonly used marker gene coding for the magnesium chelatase subunit I, involved in chlorophyll biosynthesis (Figure 1a) [24,25]. To ensure a more lasting silencing effect, we choose to introduce a sense sequence rather an inverted-repeat fragment [9]. The recombinant BMYV-SUL construct was agroinoculated to twelve *B. vulgaris* seedlings of a susceptible line further referred to as line A and plants were visually inspected over the weeks post-agroinfection. Non-agroinoculated and wild-type BMYV-agroinoculated plants were included as controls.

By ten days post-infiltration, a faint yellow vein-clearing phenotype was observed on seven out of twelve BMYV-SUL agroinoculated plants. The vein chlorosis started to appear by the third week post-infiltration and only from the third or fourth pairs of leaves, always starting from the basal section of the leaf, close to the petiole and spreading towards its apex until the whole leaf displayed the yellow vein-clearing feature (Figure 1b). As the plants developed, leaves grew larger, and the vein chlorosis became more obvious and easily noticeable. These symptoms, hereafter referred to as VIGS symptoms, are characteristic of the knockdown of *CHLI* transcripts, as previously observed in other plant models, notably *A. thaliana* [9], *Nicotiana benthamiana* [26], and *Pisum sativum* [27]. The visual screening was performed up to five weeks post-infiltration, whereby ten out the twelve plants exhibited strong VIGS symptoms while plants agroinoculated with the wild-type BMYV clone showed no visible symptom that distinguished them from non-agroinoculated plants.

The infection status of each plant was determined four weeks post-infiltration by conducting TAS-ELISA experiments on apical non-infiltrated leaves. An infection rate of 83% (10 out of 12 plants) was observed for both the wild-type BMYV and recombinant BMYV-SUL constructs. It is noteworthy that, for the BMYV-SUL condition, presence of BMYV was detected only in VIGS symptomatic plants confirming the infectivity of the recombinant virus and the effectiveness of the silencing of the target gene. The differences between infected and non-infected conditions were found to be significant for both wild-type and recombinant viruses and no significant difference was found between the two viral conditions.

To verify the stability of the insert within the modified viral genome, the progeny was analysed by RT-PCR on total RNA extracted from apical non-agroinoculated leaves of wild-type BMYV and recombinant BMYV-SUL agroinoculated plants, six weeks post-infiltration. Viral cDNAs were reverse transcribed and the 3′-end of the BMYV genome was amplified by PCR using oligonucleotides flanking the inserted SUL fragment site. A non-inoculated plant (NI) was used as negative control. As expected, amplicons of approximately 402 bp were observed only in plants inoculated with wild-type BMYV (samples 1 to 6) while in recombinant BMYV-SUL inoculated plants (samples 8 to 12), the amplicons were slightly larger, around 472 bp, indicating the presence of the SUL fragment insertion (Figure 1d). No PCR fragment was observed in plants 4 and 7, which were previously detected as negative for BMYV infection by TAS-ELISA, thereby confirming the absence of infection in these samples. Sequencing of the PCR amplicons revealed the presence of the 70-nucleotide sequence only for BMYV-SUL infected plants which shared 100% nucleotide identity with the originally inserted SUL sequence. These results confirmed the ability of the BMYV-SUL cDNA clone to establish systemic infection in sugar beets with high efficiency. Moreover, the recombinant viral clone can trigger visible symptoms in *B. vulgaris* plants and proved its stability over multiple viral replication cycles.

### 3.2. Evaluation of Recombinant BMYV-SUL Based VIGS in Different Sugar Beet Lines

Several studies evidenced that the effectiveness of VIGS and severity of visual phenotype are genotype-dependent, contingent on its susceptibility to the viral infection or genotype-specific factors [28,29,30,31,32]. To assess the efficiency of the recombinant BMYV-SUL as VIGS vector in different sugar beet lines and evaluate the genotype-response, four *B. vulgaris* lines with different susceptibilities toward BMYV, namely B, C, D and E were challenged by wild-type BMYV and recombinant BMYV-SUL agroinoculation.

By two weeks post-infiltration, yellow vein clearing symptoms could already be observed on BMYV-SUL challenged plants of line D and E, while plants of lines B and C took around three to four weeks for symptoms to manifest. Moreover, there was a striking difference in the intensity of symptoms among the different lines and in the percentage of symptomatic plants (Figure 2a). Plants of lines D and E exhibited pronounced vein clearing and yellowing across the entire leaf surface, while leaves of plants of lines B and C were mostly green, with fainter vein clearing symptoms, mainly visible in the upper laminal section of the leaves, close to the midrib. Unlike lines D and E where all the apical leaves were symptomatic, younger leaves (as from the sixth or seventh pairs) of lines B and C plants remained asymptomatic. Additionally, it is worth noting that a few leaves of plants of line C displayed a strong photobleaching phenotype, a trait which was not observed in the other three lines (Figure 2a). In this phenotypic screening, 10% (1 out of 10) of VIGS symptomatic plants was recorded for line B, 50% (5 out 10) for line C, 90% for line D (9 out of 10) and 100% for line E (Figure 2b). Five weeks post-infiltration, leaf discs were harvested from each plant to determine their infection status by TAS-ELISA. All VIGS symptomatic plants of the four lines were found to be infected by BMYV-SUL and conversely, asymptomatic plants were detected as non-infected. None of the plants agroinoculated with the wild-type BMYV clone showed any visible symptom.

We further compared the infectivity of the recombinant BMYV-SUL with the wild-type virus in the four sugar beet lines. For lines D and E, almost similar infection rates, around 80% and 100%, respectively, were found for both wild-type and recombinant BMYV conditions (Figure 2b). For lines B and C, an infection rate of 30% was found on wild-type BMYV agroinoculated plants and of 50% and 10%, respectively, on BMYV-SUL agroinoculated plants. Furthermore, no significant difference was found between the mean OD_405nm_ values recorded for the wild-type BMYV and recombinant BMYV-SUL agroinoculated plants, for all four lines (Figure 2b). While significant differences between non-infected plants and BMYV wild-type or BMYV-SUL agroinoculated plants were observed for lines D and E, lines B and C showed no significant difference among the three conditions, suggesting a lower degree of susceptibility to BMYV infection, contrary to lines D and E.

This experiment confirmed that the development of VIGS symptoms in sugar beets is genotype dependent and might be related to the level of susceptibility of the genotype to the viral infection. Moreover, ELISA results indicate that the recombinant BMYV-SUL is as infectious as the wild-type virus, and that the number of symptomatic plants is representative of the infection rate of the given population of sugar beets. Most importantly, in all lines tested, the recombinant virus BMYV-SUL showed the same tendency of infectivity as the BMYV wild-type clone. Lines D and E displayed a high infection infectivity (80 to 100%, respectively) indicating the potential of BMYV-SUL to be used as a screening tool for further studies.

### 3.3. Estimation of Viral Concentration from Transformed ELISA Absorbance Values

To better understand the relationship between sensitivity to BMYV infection and the development of VIGS symptoms, ELISA absorbance values were exploited as a semi-quantitative approach to estimate the viral load in sugar beets of different lines, under different infection conditions and at different time points post-infiltration. However, given the variability within and between ELISA assays [33], using raw OD_405nm_ values to compare the level of infection among the experiments would have been inaccurate. To improve the analysis, we included a reference standard within each microtiter plate to normalise the ELISA readout to that of the reference standard [34].

A recent study proposed a five-parameter logistic regression (5PL) model to describe the relationship between ELISA absorbance values and protein concentration, whereby predicted calibration curves of OD values against dual logarithmic protein concentration were generated using a serial dilution of twelve samples on several microtiter plates [18]. To verify this model, the same experiment was conducted in triplicate, using 4.5 ng/µL of purified BMYV and OD_405nm_ values, measured after 30 and 60 min, and used to visualise the distribution of the response variable. Although the data points did not correlate perfectly with the described model, the non-linear relationship between OD_405nm_ values and ${Log}_{2}[$viral concentration (pg/µL)] was again observed, reinforcing the applicability of this model for viral load estimation (Figure 3a).

Given the time-consuming procedure of virus purification, using purified virus as a reference standard for ELISA experiments seemed inconvenient for large-scale analysis. As a simpler and cost-effective alternative, standard samples were prepared by grinding leaves from BMYV-infected sugar beets and equally aliquoting in Precellys^®^ tubes, to be used as standard sample (see Materials and Methods, Section 2.5). To estimate the viral concentration of the standard samples, three aliquots were analysed by TAS-ELISA and the OD_405nm_ measurements were transformed to the relative viral concentrations by interpolation to their respective calibration curve, using an inverse logistic regression model [18]. In dual logarithmic form, concentrations of 9.30 pg/µL, 11.7 pg/µL, and 9.14 pg/µL were found for aliquot 1, 2, and 3, respectively (Figure 3b). The dual logarithmic concentration of the reference standard was thus approximated to 10 pg/µL and henceforth, ELISA standard curves were drawn from a 10-step serial dilution of the prepared standard samples.

Thus, to normalise ELISA measurements of independent assays, absorbance values measured at 30 and 60 min of each sample were fed into their respective calibration model and transformed by the inverse regression model. The mean of the transformed data of both time points was calculated and used as response variable to estimate the viral load of each plant [18,19]. As reported by Djaïleb et al. [35], the limitations of these models are their asymptotes. Data points below the bottom asymptote or those transformed to negative values were considered non-infected and were assigned a value of zero. Plants were considered as infected when they presented a transformed value above the mean of that of non-inoculated controls. The novelty here is that the OD values of each sample within the serial dilution were correlated to the dual logarithm of known virus concentrations and, as a result, this comprehensive data analysis approach enabled accurate comparison of the levels of infection across the different conditions.

### 3.4. Time Course Analysis of BMYV-SUL Infection and Symptom Development

Given the variation observed in the onset of VIGS symptoms among the different *B. vulgaris* lines, it seemed relevant to further investigate the establishment of viral infection and development of VIGS symptoms over a period of time. The aim was to determine the appropriate time interval, after infiltration of the BMYV-SUL clone for phenotypic recording of sugar beets. Thus, a time course experiment was conducted on two lines having previously demonstrated a diverging degree of susceptibility to BMYV wild-type and BMYV-SUL infection, namely lines B and E. Plants of each line were agroinoculated with the wild-type BMYV and recombinant BMYV-SUL clones and over a period of eight weeks, each plant was visually inspected every two weeks. Simultaneously, discs were harvested from newly developed leaves to determine the infection status of the plants by TAS-ELISA analysis.

Two weeks post-infiltration, faint VIGS symptoms were observed on two of the ten BMYV-SUL agroinoculated plants of the line B. By the sixth week, two additional plants exhibited symptoms, bringing the total to 40%, and these symptoms persisted till at least the eighth week post-infiltration (Figure 4a). For line E, strong VIGS symptoms were visualised on new leaves for eight out of the ten BMYV-SUL agroinoculated plants as early as two weeks post-infiltration and, by the eighth week, all the plants were symptomatic. Thus, the visual analysis showed a progressive increase in the percentage of plants with VIGS symptoms over time for both lines.

To investigate whether the increase in symptom-exhibiting plants correlated with viral accumulation, the ELISA absorbance values obtained for all plants were normalised and transformed according to their calibration curves as detailed in Section 2.7 and Section 3.3. Only data above the mean of the transformed values from non-inoculated controls were used to assess viral loads upon infection conditions. Contrary to the phenotypic evaluation, both lines showed similar concentration curves with a gradual decrease in viral titers starting between four and six weeks post-infiltration for both wild-type and recombinant BMYV (Figure 4b). Plants of line E consistently exhibited higher infection levels than those of line B. Interestingly, some plants, particularly of line B, began to display VIGS symptoms at a time point when the viral load was estimated close to null. This analysis suggests that the onset of VIGS symptoms is triggered with some delay and that in line B, another parameter may play a role in addition to the virus titer.

Based on these results, it appears that the time interval necessary for the multiplication of the virus, its induction of silencing and eventually the development of VIGS symptoms, varies greatly depending on the susceptibility level of the genotype to BMYV infection. Therefore, it seems delicate to set a standard time interval for phenotypic screening in studies using BMYV-SUL and it may be necessary to adjust the screening schedule according to the given genotype for efficient assessment of VIGS.

### 3.5. Analysis of Viral and Small Interfering RNAs in BMYV-SUL-Infected Plants

Considering the disparity observed between viral infection and the onset of silencing symptoms, a molecular analysis of both viral genomic and small RNAs was undertaken to better understand the temporal establishment of BMYV infection, its induction of silencing of the SUL gene, and the subsequent development of symptoms in *B. vulgaris.* We reproduced the former experiment with sugar beets of lines B and E agroinoculated with wild-type BMYV and recombinant BMYV-SUL constructs. The plants were analysed by monitoring visually the appearance of the VIGS symptoms and assessing the viral concentration by TAS-ELISA and the viral RNA by northern blotting at two and six weeks post-infiltration.

By two weeks post-infiltration, faint yellow vein clearing was observed on the third pair of leaves of one of the ten agroinoculated plants of line B. Although the growth rate of the line E was slower, vein-clearing symptoms were observed, or for some plants, suspected on the developing third pair of leaves for eight of the ten agroinoculated plants (Figure 5a,b). At six weeks post-infiltration, three more plants of line B exhibited vein chlorotic symptoms, spreading across the entire leaf surface, while for line E, all plants were strongly symptomatic. Consistent with previous observations, the number of symptomatic plants increased over time and a global decrease in the viral titer was found (Figure 5b). The decrease was found to be significant for both viral constructs in line E. In the case of the line B, statistical analysis could not be performed due to the low number of infected plants.

The decrease in viral load from two to six weeks post-infiltration was further investigated by analysing genomic and subgenomic viral RNAs of infected plants. Total RNAs were individually extracted from non-inoculated, wild-type BMYV or recombinant BMYV-SUL infected plants, harvested at both time points and analysed by northern blot, using a radioactive-labelled DNA probe targeting both genomic and subgenomic BMYV RNAs (Figure 5c). The estimated dual logarithmic viral concentrations of each sample are indicated as a histogram, and asymptomatic and symptomatic plants are respectively annotated by a minus (−) or a plus (+) sign. For plants of line B, of the four plants infected with BMYV-SUL and analysed by northern blot, only one symptomatic plant (number 8) showed a strong signal indicating the presence of genomic and subgenomic viral RNAs at two weeks post-infiltration. Three out of the four plants inoculated with wild-type BMYV and analysed reacted positively (Figure 5c, left panel). At six weeks post-agroinfiltration, viral RNAs were detected in all plants, except in plant (1) inoculated with wild-type BMYV for which no RNA loading was detected. At this time point, the previously negative plants inoculated with BMYV-SUL (5–7) were all symptomatic. For line E, as most plants inoculated with BMYV-SUL were symptomatic at two weeks post-inoculation, six plants out of the ten inoculated were analysed. Viral RNAs were observed for all symptomatic plants at both time points, including the wild-type BMYV agroinoculated plants, except plant (2) at two weeks post-infiltration (Figure 5c, right panel). For plants (3), (4), and (6) inoculated with BMYV-SUL, viral RNAs were weakly detected due to a very low RNA loading and concentration.

To further investigate the discrepancy between development of symptoms and gene silencing occurrence, we analysed the presence of SUL-specific siRNAs by northern blot, using a probe homologous to the 70 nts SUL insert. RNA loading was controlled by hybridising the blot with a universal probe corresponding to the 21-nucleotides miR160. As expected, no signal was detected for non-inoculated and wild-type BMYV agroinoculated plants, for either line (Figure 5d). At two weeks post-infiltration, the only plant of line B (plant (8)) showed two bands corresponding to the 21–24 nucleotides siRNA, attesting the presence of SUL siRNA. This plant was the only one which displayed VIGS symptoms and showed a strong signal for genomic and subgenomic RNAs. At six weeks post-infiltration, siRNA signals were detected for only plants (5), (6), and (7), while VIGS symptoms were observed on the four plants analysed (Figure 5d). For line E, siRNA signals were detected for the six BMYV-SUL agroinoculated plants (3) to (8) which displayed VIGS symptoms already at two weeks post-infiltration. Four weeks later, siRNAs from plant (3) (very faint), (5), (6), and (7) could still be detected. Results corresponding to plants (4) and (8) at six weeks post-infiltration could not be interpreted due to the absence of small RNAs on the blot (no miR160 detected) (Figure 5d). This experiment showed that the presence of vein chlorosis correlated with a high accumulation of viral RNA, and thus also the presence of siRNAs, irrespective of the time point. This was particularly true for plants of line E that accumulate higher virus titers. Conversely in line B, although all inoculated plants were positive by RT-PCR (Appendix A), these plants contained less virus (Figure 5b,c) and therefore induced less VIGS symptoms both at early and later time points (one and four positive plants, respectively).

### 3.6. Persistence of VIGS Symptoms in Sugar Beets

VIGS has long been considered a transient assay system, generally lasting for three to four weeks as the plant starts to recover from the gene silencing [36,37]. Nonetheless, recent studies have reported that in some plant pathosystems, VIGS can persist for several years and potentially until the plant dies [38,39]. In this study, sugar beets were not maintained for such extended periods, but we found that up to five months post-agroinoculation, BMYV-SUL infected plants continue to exhibit silencing symptoms. However, upon close inspection of the youngest leaves, VIGS symptoms were always observed earlier for line E compared to line B (Figure 6a), and by five months, they were no longer detected in line B.

To verify whether the disappearance of silencing symptoms in new leaves could be related to absence of infection or reversion of the recombinant BMYV-SUL to its wild-type state, leaf discs were randomly harvested from newly developed leaves at 14 weeks post-infiltration and analysed by TAS-ELISA and RT-PCR. By comparing the mean transformed values obtained 14 weeks post-infiltration to those measured at two and six weeks post-infection, a strong decrease in viral load was marked for both lines (Appendix A). The presence of the SUL insert in the viral genome progeny was analysed by performing RT-PCR on RNA extracts from the youngest leaves of plants, previously diagnosed as infected. Except for plant 7 of line B, all agroinoculated plants were detected as positive to BMYV infection and the PCR bands corresponded to the expected sizes of the respective agroinoculated clones, regardless of the presence or absence of silencing symptoms (Figure 6b). These results bolster the stability of the BMYV-SUL virus, showing that the insert is retained over multiple replication cycles, even months after infection, and suggest that persistence of VIGS is genotype-dependent.

## 4. Discussion

The aim of this study was to develop a molecular tool that enables breeders to visually identify and discard sugar beet varieties susceptible to polerovirus from large accessions, allowing breeding programs to further focus exclusively on potentially resistant varieties. To date, only one analogous approach has been described in the literature for sugar beet. By labelling the potyvirus BtMV with the *B. vulgaris* transcription factor MYB1, the team of Rollwage et al. [40] was able to visualise with the naked eye the systemic spread of the virus as red pigmentation emerging across sugar beet leaves. Unfortunately, due to their compact genome organisation and packaging constraints [4,5,6], addition of large sequences such as the BtMYB1 gene (675 nucleotides) into the polerovirus genome is likely to be deleterious or detrimental to their infectivity and stability. In this work, we propose an alternative visual screening tool applicable to polerovirus infection in sugar beets.

We previously reported that polerovirus-derived VIGS vector can effectively induce silencing of endogenous host genes in the plant model *A. thaliana*. In this study, we adapted this approach to the sugar beet-infecting polerovirus BMYV, by adding a 70 bp fragment of the endogenous *B. vulgaris* marker gene *CHLI* (SUL) at the 3′ UTR of the viral cDNA genome. Following syringe agroinoculation of the recombinant BMYV-SUL, visual vein clearing of leaves characteristic of CHLI silencing appeared, as early as two weeks post-infiltration, usually as from the third pair of sugar beet plants, while only the cotyledons and first pair of leaves were infiltrated. The effectiveness of the BMYV-SUL construct was evaluated in five genetically different *B. vulgaris* lines, namely A, B, C, D, and E. Genotype-dependent variations in VIGS symptoms, ranging from faint vein-clearing photobleaching or strong vein clearing and yellowing of leaves, were observed, similar to previous VIGS studies carried out in diverse agricultural crops [28,29,30,32,41]. Molecular diagnosis revealed that, despite the varying susceptibilities to viral infection among the five lines, both the wild-type and recombinant virus consistently exhibited similar infectivity, indicating that the insert did not introduce any bias in pathogenicity. Moreover, a perfect correlation between percentage of VIGS symptoms and BMYV infection rates was found, supporting that VIGS effectiveness is dependent on the susceptibility of the genotype to the viral infection [42,43]. For the least susceptible line B, the VIGS effectiveness of BMYV-SUL fluctuates between 10% and 40%, while for the most susceptible line E, an effectiveness of 100% was achieved. We also observed that the onset of symptoms was clearly delayed in the least susceptible line compared to the susceptible line.

Since VIGS symptoms appeared at variable time points across the different sugar beet lines, we attempted to determine the optimal time interval for accurately evaluating BMYV susceptibility through visual screening. The time course analyses of recombinant BMYV-SUL infection and silencing indicated that onset of symptoms is correlated to the genotype’s susceptibility to the infection. While two weeks was sufficient to visually identify infected plants of the highly susceptible line, VIGS symptoms could take up to six weeks to appear in infected plants of the less susceptible line. Our findings report that, similar to tobacco mosaic virus (TMV)-based VIGS in *N. benthamiana* [44] or potato virus X (PVX) in potato [45], the onset of visual VIGS symptoms relies on a prerequisite threshold of viral RNAs in the host plant. The effectiveness of the BMYV-SUL as VIGS vector depends on its ability to invade the host plant, establish its replication cycles, and accumulate until a sufficient pool of dsRNA molecules is generated to induce the PTGS pathway. This subsequently triggers siRNAs production, which in turn silence the *CHLI* gene, ultimately instigating the appearance of visible symptoms. It has been demonstrated for citrus leaf blotch virus (CLBV)-based VIGS that the required viral accumulation threshold varies among plants of different species [46]. It would be interesting to verify, by using a more sensitive technique like real-time quantitative PCR, whether the threshold required to trigger symptoms is similar in all sugar beet lines or varies between genotypes.

Despite unforeseen disparities were observed in symptom induction among lines, this approach allows to discard highly susceptible lines within only two weeks post-infection. Importantly, it remains more reliable and time efficient compared to scoring yellowing symptoms, typically appearing from at least five to eight weeks and are not specific to viral infection [1]. The efficiency of VIGS as a rapid large-scale screening tool for plant resistances has been recently reported for the cassava mosaic disease (CMD) [47]. By engineering the east african cassava mosaic virus (EACMV) as a vector to induce silencing of the endogenous gene SPINDLY (SPY) gene, known to encode a negative regulator of the gibberellin signalling pathway [48], CMD-susceptible cassava lines could be easily distinguished from resistant ones, based on induced shoot-tip necrosis and eventual plant death occurring only in susceptible lines. Potentially resistant cultivars were identified within four weeks post-infection, which is eighteen weeks faster than traditional symptom scoring methods.

VIGS is generally considered as a transient assay system, efficient for a limited time. In sugar beets, a progressive reduction in virus accumulation was observed and viral replication persisted at a low level, up to at least five months post-infiltration, irrespective of the susceptibility of the line to BMYV. Interestingly, the host insert SUL of 70 nucleotides (in a sense orientation) was sufficient to trigger RNA silencing and was integrally maintained in the viral progeny even over months of infection. This highlights the remarkable stability of the recombinant BMYV in the sugar beet host and the robustness of the technology. This feature was already observed in *A. thaliana* where *CHLI* sense sequences of 42 and 81 nts inserted into the genome of turnip yellows virus (TuYV) proved to be stable while a hairpin insert of 108 nts did not [9]. Nevertheless, despite the BMYV-SUL infection, VIGS symptoms were hardly visible on developing leaves of the less susceptible line. A possible explanation could be that BMYV, along with the *CHLI* gene, was silenced, leading to a significant reduction in viral replication and dsRNA levels, ultimately resulting in symptom recovery, as previously observed for TMV- and tobacco rattle virus (TRV)-derived VIGS experiments in *N. benthamiana* [36,49].

In contrast, in the highly susceptible line, VIGS symptoms persisted in all plants, with typical yellow vein chlorosis visible across the leaf surface and in all leaves, even months after infection. Interestingly, persistence of VIGS with low levels of virus, lasting for more than three years, have been reported for CLBV-based VIGS in citrus [46] and for more than twelve months for TRV-based VIGS in tomato and *N. benthamiana* [38]. Although the underlying mechanism for VIGS’ long-term persistence is not fully understood, our hypothesis is that despite reduced viral accumulation over months post-inoculation, enough replication cycles are maintained in susceptible lines, thereby producing sufficient dsRNAs to trigger CHLI silencing in new developing leaves. Further quantitative studies should be undertaken to investigate the temporal and spatial dynamics of both viral RNA and siRNA accumulation, siRNAs movement, and silencing in tolerant and susceptible sugar beet lines.

With the advent of molecular biotechnology, VIGS has emerged as a highly efficient and economical forward genetic screening technique in a variety of agronomically important non-model plants [11,50,51]. It is particularly valuable for studying gene functions involved in resistance to diseases in crop plants [52] and one example is the identification of the plant immune signalling-related kinase, MKK6, as a positive regulator in the defence against potato virus Y (PVY) [43]. This was demonstrated by an increase in the susceptibility of different wild potato relatives to PVY infection as a result of silencing of the endogenous MKK6 gene using TRV-based VIGS. Though primarily described as a screening tool for identifying infected plants, BMYV could similarly be utilised as a VIGS vector for genomic functional analysis in sugar beets. By leveraging transcriptomic data, genes that are upregulated during viral infections could be silenced and studied in sugar beet lines of varying levels of susceptibility to BMYV using the VIGS technique. By replacing the *CHLI* insert in the BMYV-SUL construct with the sequence of the target gene, specific genes, for instance those encoding potential resistance proteins, could be silenced. Analysis of viral accumulation in those lines could contribute to identify factors involved in plant immunity.

Given the long reproduction cycle of *B. vulgaris* and genome complexity, generating transgenic lines can be time consuming and difficult [53,54]. A major advantage of VIGS-based studies is that this method circumvents the need for plant transformation, especially for crops that are recalcitrant to transformation [55,56]. BMYV-based VIGS can also be used as a functional genomic tool in the identification of novel traits involved in other pathogen defence response, abiotic stress such as drought, temperature or salinity, or genes involved in developmental processes in *B. vulgaris* [57].

In the case of high-throughput studies, one limitation in our proposed protocol can be the laborious and time-consuming process of syringe agroinoculation for viral vector delivery [10]. Nevertheless, vector transmission experiments with the polerovirus TuYV have demonstrated that the CHLI-labelled polerovirus can be efficiently transmitted by its aphid vector *Myzus persicae*, while still retaining its ability to induce VIGS symptoms [9]. This finding is promising for developing aphid-mediated BMYV-SUL infections, which would be far more compatible for larger scale applications.

## 5. Conclusions

In short, we have developed a rapid and reliable VIGS-based screening tool applicable for sugar beet crop. By inducing characteristic symptoms in the leaves, this tool enables easy visual discrimination between sugar beet lines that are resistant or tolerant to polerovirus from those that are susceptible. The high-throughput application of this tool would allow breeders to streamline pre-breeding programs and concentrate on developing high-performing sugar beet varieties capable of withstanding VY outbreaks. Altogether, our polerovirus-based VIGS system can be transposed as a useful screening tool for other viral species and other crops whether in pre-breeding programs or conventional functional genomics studies.

## Figures and Tables

**Figure 1 viruses-16-01823-f001:**
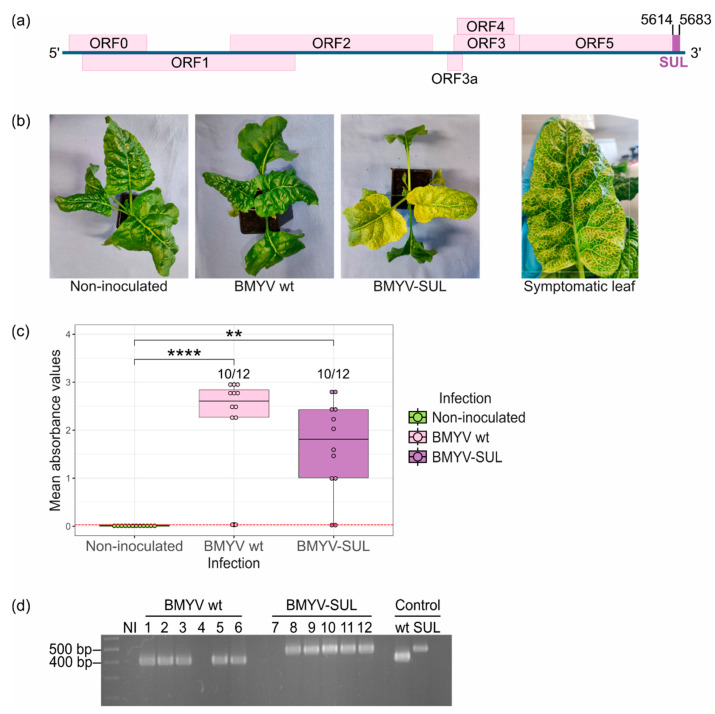
Infectivity assay of agrobacterium BMYV-SUL clone in *B. vulgaris* plants (line A). (**a**) Schematic representation of BMYV-based VIGS vector construction. A 70 nts long fragment of the endogenous *B. vulgaris* SUL gene, shown as a purple rectangle, was inserted at position 5614 of the viral genome. (**b**) Symptoms of vein clearing and leaf yellowing observed on sugar beets agroinoculated with the recombinant BMYV-SUL clone three weeks post-infiltration. No symptoms were observed on non-agroinoculated plants or plants which were agroinoculated with the wild-type (wt) BMYV clone. (**c**) Virus accumulation analysed by TAS-ELISA. Each dot of the boxplots represents the absorbance value recorded for each plant. Values from non-inoculated samples are shown in green, of wild-type BMYV agroinoculated plants in pink, and of recombinant BMYV-SUL in purple. The red dotted line indicates the positivity threshold. Statistical analyses were performed using a Kruskal–Wallis test followed by a Dunn’s test for pairwise comparisons and Bonferroni correction. The stars above the bars indicate the statistically significant differences in absorbance values for the three infection conditions. *p*-values less than 0.01 and 0.0001 are respectively flagged (**) and (****). (**d**) Stability of the SUL insert verified by RT-PCR. The first lane corresponds to the 10 kb MassRuler^TM^ DNA ladder, ‘NI’ refers to non-agroinoculated samples, lanes 1 to 6 correspond to the wild-type BMYV agroinoculated samples, and lanes 7 to 12 correspond to recombinant BMYV-SUL samples. Plasmid DNA of the wild-type and recombinant BMYV constructs were used as PCR size controls, as shown in the last two lanes.

**Figure 2 viruses-16-01823-f002:**
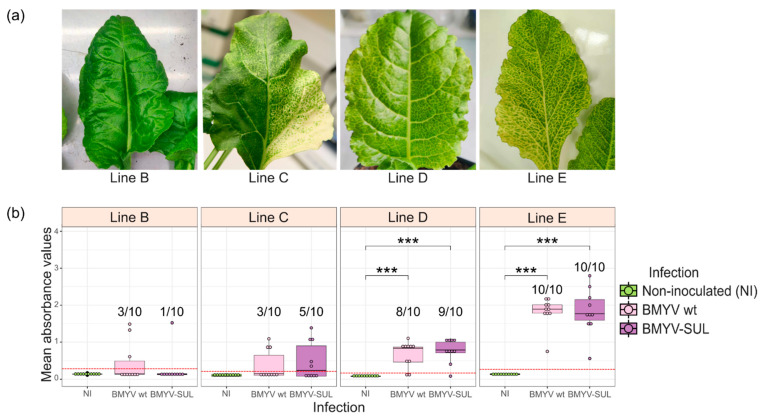
BMYV-based VIGS experiment in four different sugar beet lines (B–E). (**a**) Variation in intensities of silencing phenotypes observed in each line. (**b**) BMYV titer analysed by TAS-ELISA. The absorbance readings are displayed as box plots with non-agroinoculated (NI) plants in green, wild-type BMYV agroinoculated plants in pink and recombinant BMYV-SUL agroinoculated plants in purple. Points above the red dotted line are considered as infected and the infection rates are indicated on the graph. Statistical analyses were performed using a Kruskal–Wallis followed by a Dunn’s test for pairwise comparisons and Bonferroni correction. The stars above the bars indicate the statistically significant differences, *p*-values less than 0.001 are flagged with (***), and absence of star indicates no statistically significant difference (*p* > 0.05).

**Figure 3 viruses-16-01823-f003:**
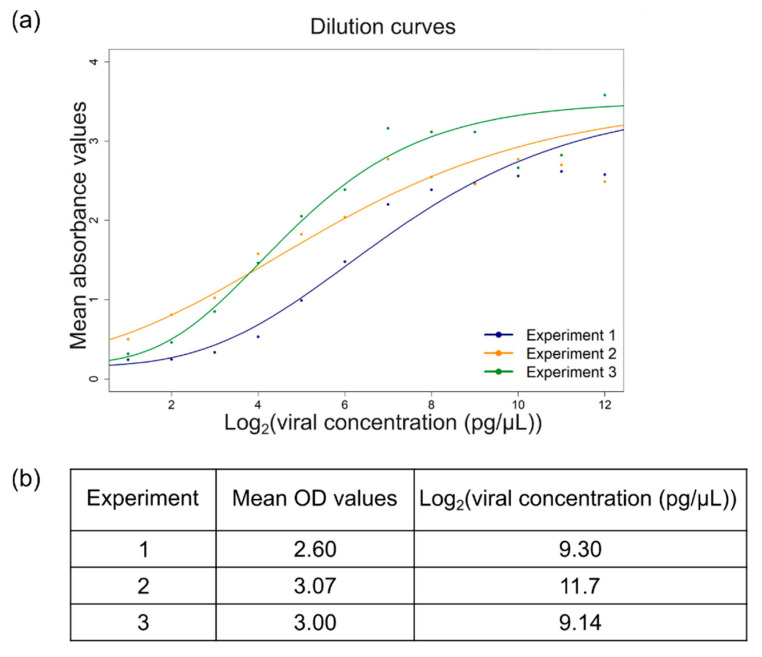
Statistical modelling of the relationship between ELISA absorbance values and viral concentrations. (**a**) Three serial dilution curves of purified virus generated in green, orange, and blue, using a five-parameter logistic regression model, following the script provided by Lange et al. [18]. (**b**) Viral concentrations of the standard sample estimated by interpolating measured absorbance value against their respective calibration curve.

**Figure 4 viruses-16-01823-f004:**
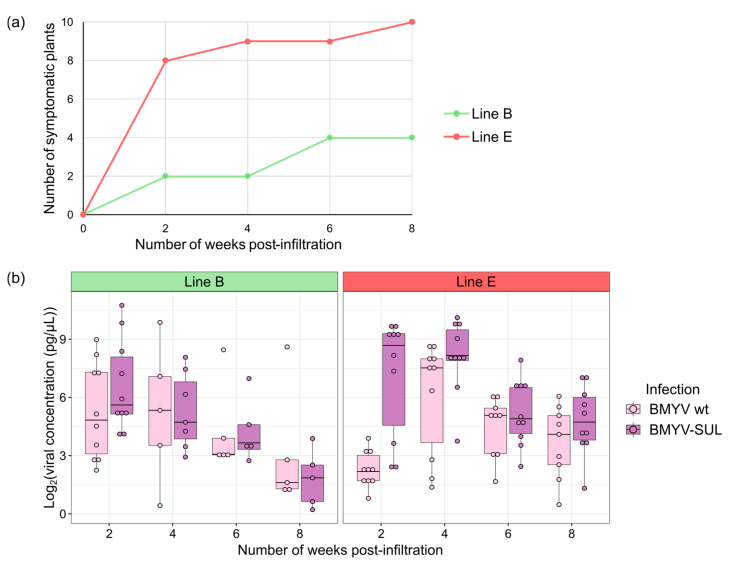
Time course of BMYV-SUL infection in sugar beet lines B (in green) and E (in red). (**a**) Visual evaluation of sugar beet symptoms agroinoculated with BMYV-SUL constructs every two weeks, over a period of eight weeks. The number of plants displaying the typical silencing phenotype at each time point is represented. (**b**) Viral accumulation in plants agroinoculated with the wild-type BMYV or recombinant BMYV-SUL constructs, estimated for each time point by transforming ELISA absorbance, based on the reference dilution curve. The points above the average of transformed values of non-inoculated controls are displayed as boxplots.

**Figure 5 viruses-16-01823-f005:**
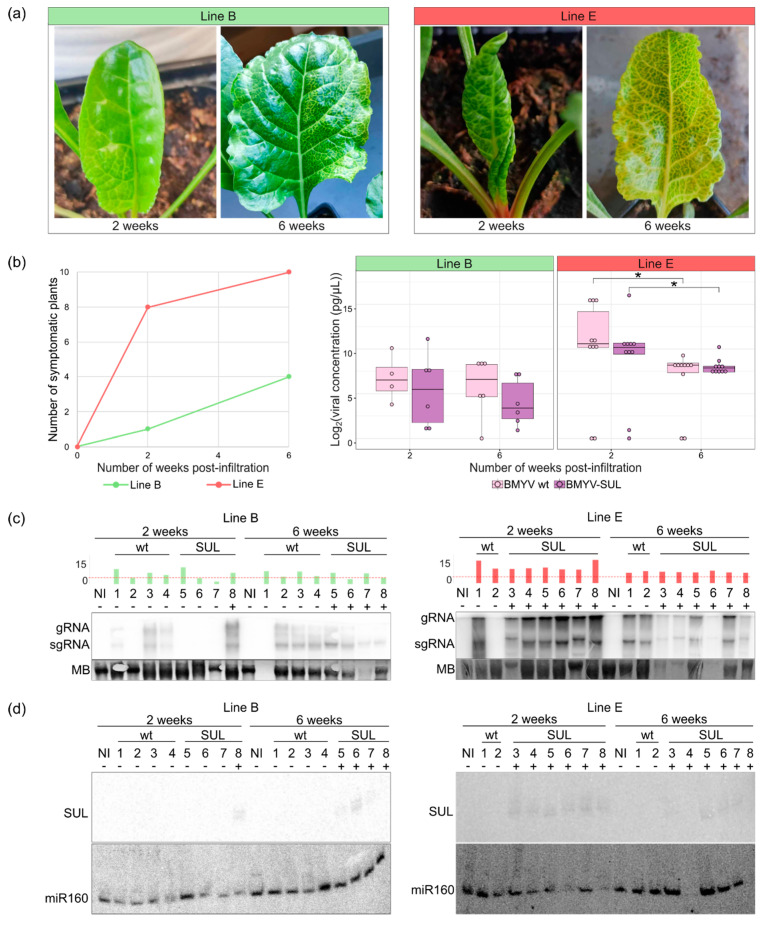
Comparative analyses of BMYV-SUL infection over time in two sugar beet lines B and E. (**a**) Symptoms development of sugar beets of the line B (**left**) and line E (**right**) challenged with BMYV-SUL at two and six weeks post-infiltration. (**b**) Number of symptomatic plants of each line, green for plants of line B and red for those of line E (**left** panel). The boxplot analysis (**right** panel) shows the dual logarithmic viral load of the plants of each line for the wild-type BMYV (in pink) and recombinant BMYV-SUL (in purple) at two and six weeks. Bars with a star indicate statistically significantly differences (*p*-value < 0.05), according to the pairwise Kruskal–Wallis and Dunn’s comparisons tests. (**c**) Northern blot analysis of viral accumulation in non-inoculated (NI), wild-type (wt) or recombinant (SUL) BMYV agroinoculated plants of line B (**left**) and line E (**right**). Genomic (gRNA) and subgenomic RNA (sgRNA) of BMYV were detected by using radioactive labelled DNA probe against the P3P5 coding sequence. Methylene blue (MB) staining was used to verify RNA loading. The dual logarithmic viral load of each plant, estimated by TAS-ELISA, is shown above the northern blots as green bars for line B and red for line E, and a red line is drawn at Log_2_(viral concentration) equals to 4 pg/µL. Asymptomatic and symptomatic plants are respectively annotated by a minus (−) or a plus (+) sign. (**d**) siRNA detection by northern blot at 2 and 6 weeks, using radioactive-labelled DNA probe targeting the SUL sequence. A universal 21-nucleotides miR160 probe was used to verify loading.

**Figure 6 viruses-16-01823-f006:**
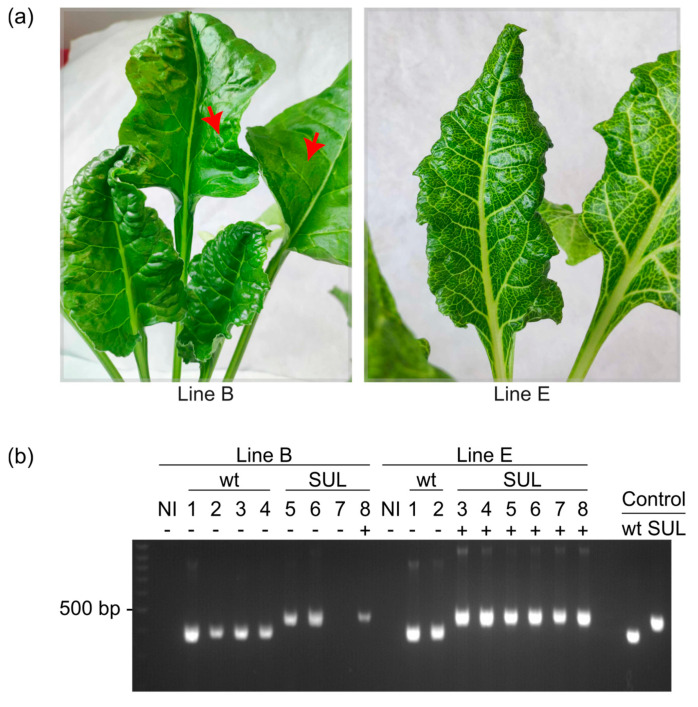
Persistence of BMYV-SUL-induced VIGS in sugar beets at 14 weeks post-infiltration. (**a**) Young leaves of symptomatic BMYV-SUL-infected beets of line B (**left**) and line E (**right**). The red arrows indicate the faint symptoms present on older non-inoculated leaves of line B. (**b**) Analysis of the infection status of agroinoculated plants of both lines by RT-PCR, using oligonucleotides flanking the inserted SUL fragment site. Bands of 402 bp and 472 bp were respectively expected for wild-type BMYV and recombinant BMYV-SUL constructs. Plasmid DNA of the wild-type and recombinant BMYV constructs were used as PCR size controls. Asymptomatic and symptomatic plants are respectively annotated by a minus (−) or a plus (+) sign.

## Data Availability

The data presented in this study are available on request from the corresponding author.

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
