# Peer review of "Rapid and Visual Screening of Virus Infection in Sugar Beets Through Polerovirus-Induced Gene Silencing"

_viruses, 2024, doi:10.3390/v16121823_

Round 1

Reviewer 1 Report

Comments and Suggestions for Authors

The authors, Heemee Devi Bunwaree and co-workers, presented a manuscript entitled, "Development of polerovirus-induced gene silencing for rapid and visual screening in sugar beets".

They created a recombinant beet mild yellowing virus (BMYV) that develops clear symptoms of virus infection, allowing early evaluation of sugar beet genotypes' resistance to the virus. They used the agroinfiltration technique for experimental inoculation of wild-type and recombinant viruses.

The paper is well written and the procedures and results are described in great detail.

The discussion compares the results obtained and highlights possible aspects of their findings for future development and breeding of sugar beet.

I can recommend the manuscript for publication in the journal Viruses after some minor issues have been addressed.

Line 152: anti-BWYV antibodies – explain the relationship to BMYV. Also briefly describe the TAS-ELISA procedure used.

Line 157 – threshold for ELISA positive samples should be determined by a more appropriate method (e.g. mean of negative control plus 3 SD)

Chapter 3.4. Time course analysis of BMYV-SUL infection and symptom development.     The authors used the number of plants with symptoms to express the phenotype of the effect of the recombinant virus. It would be better to express the intensity and severity of symptoms using a descriptive scale.

Comments on the Quality of English Language

A revision of the English is needed.

Reviewer 2 Report

Comments and Suggestions for Authors

It seems to me that most, if not all, analyses for VIGS status, viral replication levels, etc. could be more efficiently screened using qRT-PCR that would generate more definitive values. I don't understand the rationale for using TAS-ELISA vs. qRT-PCR as the authors state the case for using qRT-PCR (lines 627-629) to look at one aspect of the experiments. In terms of time vs effort, qRT-PCR provides more flexibility to examine different aspects of the entire system using the same extracted RNA/synthesized cDNA (plant or viral RNA detection). Can you explain why TAS-ELISA was used vs qRT-PCR?

While I have issues with the experimental approach chosen, I have no problem with the significance/usefulness of utilizing VIGS to easily test for viral susceptibility in various sugar beet cultivars. I just think the findings would be more clear cut if using qRT-PCR.

Reviewer 3 Report

Comments and Suggestions for Authors

This manuscript describes development of VIGS system in sugar beet using beet mild yellowing virus (BMYV). Experiments were carried out thoroughly and the manuscript is generally well written and prepared. I suggest some points need to be clarified/added in the texts to improve the article.

-       -  Is BMYV infection always asymptomatic? As shown in Fig 1b. In the results presented in result section 3.2. it is not described regarding the viral symptoms associated with BMYV wild-type infection.

-       -  More information should be added in the introduction section regarding the biology of BMYV such as pathogenicity and effect on the crop yield.

-        - Is the phenotypic symptom induced by BMYV-SUL infection well correlated with virus infection? For example, in the results presented in Fig. 5b, it is not clearly described the result of RT-PCR detection, although it mentioned briefly in line 542-545 that all cultivar B plants were infected, while only 4 plant showed symptoms.

-      -  It is also necessary to clarify whether symptom severity induced by BMYV-SUL is correlated with virus titer.  As presented in Fig. 5c-d, it is necessary to provide direct comparison on virus titer/RNA level between cultivar B and E. It is better to provide disease index based on visual observation and whether this correlated with virus titer.

-       -  I suggest the author to improve the blot analysis presented in Fig. 5c, particularly because the RNA loading amount is not uniform, so that it is difficult to assess the virus RNA accumulation level.

-      -   In the abstract, “As the symptoms induced often do not correlate with presence of virus”, please improve this sentence for better clarity.

Reviewer 4 Report

Comments and Suggestions for Authors

Review of the manuscript Development of polerovirus-induced gene silencing for rapid and visual screening in sugar beets

 The MS brings interesting results that can be used in practice. Altogether, I liked the text very much.

I have minor comments only:

Title: perhaps it could be more precise, eg. … screening of virus infection…

For me, it is a little bit strange that the title is not capitalized whereas titles in references are. It's usually the other way around.

L. 37: In my opinion, BtMV should not be included among viruses causing virus yellows even when it frequently occurs together with these viruses, since it causes quite different symptoms.

The authors probably used the definition from the article of Hossain et al.

L. 38-40: English names of viruses are no more considered as scientific and thus they should not be written in italics and capitalized first word. Moreover, later in the text (eg. l. 651 and 656) the names are written normally.

L. 114: Latin names of organisms should be written differently from other text. So, if the heading text is in italics, Beta vulgaris should be written normally.

L. 127: redundant full stop.

Figures: Altogether, I liked graphic differentiation of different forms of virus and sugar beet lines, but in fig. 1a SUL is not well visible because of small contrast between the two colours used. More different colours could be better. However, it would also mean changing of colours on all figures.

I miss fig. 6c (both the fig. itself and the description in the legend).

Round 2

Reviewer 3 Report

Comments and Suggestions for Authors

The comments have been sufficiently addressed